# The Potential Role of bZIP55/65 in Nitrogen Uptake and Utilization in Cucumber Is Revealed via bZIP Gene Family Characterization

**DOI:** 10.3390/plants12183228

**Published:** 2023-09-11

**Authors:** Bing Hua, Fei Liang, Wenyan Zhang, Dan Qiao, Peiqi Wang, Haofeng Teng, Zhiping Zhang, Jiexia Liu, Minmin Miao

**Affiliations:** 1College of Horticulture and Landscape Architecture, Yangzhou University, Yangzhou 225009, China; binghua@yzu.edu.com (B.H.); 15805235039@163.com (F.L.); z15095786059@163.com (W.Z.); qiaod7201@163.com (D.Q.); m15252577369@163.com (P.W.); thf2193228179@163.com (H.T.); zhangzp@yzu.edu.cn (Z.Z.); 008302@yzu.edu.cn (J.L.); 2Joint International Research Laboratory of Agriculture and Agri-Product Safety of Ministry of Education of China, Yangzhou University, Yangzhou 225009, China; 3Key Laboratory of Plant Functional Genomics of the Ministry of Education/Jiangsu Key Laboratory of Crop Genomics and Molecular Breeding, Yangzhou University, Yangzhou 225009, China

**Keywords:** cucumber, bZIP, nitrogen, NPF, nitrate transporter

## Abstract

The bZIP (basic leucine zipper) proteins play crucial roles in various biological functions. Nitrogen (N) is an essential element for plant growth, especially in cucumber (*Cucumis sativus*) due to its shallow roots. However, the regulation of bZIP genes in cucumber nitrogen metabolism has not been studied yet. In this study, we identified a total of 72 bZIP genes (CsbZIPs) in the cucumber genome that could be classified into 13 groups. These genes were unevenly distributed on seven chromosomes, and synteny analysis showed that the CsbZIP genes were expanded in a segmentally duplicating manner. Furthermore, our genome-wide expression analysis suggested that CsbZIP genes had different patterns and that five CsbZIP genes were regulated by nitrogen treatment in both leaves and roots. Consistent with CsNPF, CsbZIP55 and CsbZIP65 were regulated by nitrogen treatment in leaves and roots. Moreover, the subcellular localization showed that CsbZIP55 and CsbZIP65 were specifically located in the nucleus, and the transcriptional activation assay showed that CsbZIP55 and CsbZIP65 have transcriptional activation activity. Additionally, in the *CsbZIP55* and *CsbZIP65* overexpression plants, most nitrogen-regulated CsNPF genes were downregulated. Taken together, our comprehensive analysis of the bZIP gene family lays a foundation for understanding the molecular and physiological functions of CsbZIPs.

## 1. Introduction

Cucumber (*Cucumis sativus* L.), a horticultural crop widely cultivated worldwide, requires nutrients during its growth cycle just like other plants [1]. Among those nutrients, nitrogen is the main limiting nutrient for cucumber yield. Because of its shallow root system, cucumber depends much more on nitrogenous fertilizer than many other horticultural crops [2,3,4,5]. Due to the large amount of nitrogen lost via nitrogen leaching in the soil, the utilization rate of nitrogen is quite low at present. To increase cucumber yields, nitrogenous fertilizer is typically applied in a manner that far exceeds the requirements [6,7]. Therefore, improving the nitrogen use efficiency (NUE) and reducing nitrogen fertilizer use in cucumber is a critical challenge that is urgent to address. To date, several studies have attempted to analyse the underlying mechanism of nitrogen absorption in cucumber.

In contrast to the other three essential life elements (carbon, hydrogen, and oxygen), nitrogen must be absorbed via the root system of plants. There are various nitrogen sources in soil, such as urea, nitrate, amino acids, and ammonium. Among those nitrogen resources, nitrate is the main nitrogen resource in soil and the major nitrogen form absorbed in aerobic environments [8]. In plants, the slow anion channel-associated homologues (SLAC/SLAH), chloride channel family (CLC), nitrate transporter 2 (NRT2), and nitrate transporter 1/peptide transporter family (NPF) are the four gene families involved in nitrate transport [8,9,10]. Among those four families, the NPF family is the largest family and has important functions in nitrogen utilization. To date, the NPF family has been identified in many crops, such as *Arabidopsis thaliana* (53 members), *Oryza sativa* (74 members), *Triticum aestivum* (331 members), and *Zea mays* (79 members) [11,12,13]. In Arabidopsis, several significant functions of NPF genes have been revealed, and NPF genes have important and significant functions in nitrogen utilization. AtNPF6.3 (CHL1/AtNRT1.1), a high- and low-substrate-affinity nitrate transporter, is expressed predominantly in roots and regulated by nitrogen status [14]. In addition to the function of nitrogen absorption, AtNPF6.3 also participates in the early nitrate signalling of the primary nitrate response as a nitrate sensor [15]. Recently, the NPF gene family was identified and characterized in cucumber, and 54 NPF genes were identified [16]. Interestingly, most nitrogen-regulated NPFs were upregulated in roots, whereas most nitrogen-regulated NPFs were downregulated in leaves [16]. Despite this progress, nitrogen absorption and distribution are still ambiguous.

A set of genes have been Identified to regulate nitrogen uptake and distribution. Among them, the basic leucine zipper (bZIP) transcription factors are involved in the stress response, environmental signalling and plant development, including nitrogen utilization [17]. For instance, TabZIP60 regulates wheat nitrogen use and growth via the negative regulation of TaNADH-GOGAT expression [5]. Interestingly, TaNRT2.1, a known nitrate-inducible nitrate transporter, and TabZIP60 are negatively correlated [5]. The A, D, H, and S Group bZIP members are involved in the regulation of nitrogen use. AtbZIP1, an S Group member, propagates nitrogen nutrient signals as a master regulator [18]. AtTGA1 and AtTGA4, two D Group members, are involved in the nitrate response [19]. AtHY5 and AtHYH, two H Group members, regulate the expression of nitrate and ammonium transporters and nitrite and nitrate reductase [20,21]. AtABI5, an A group member, inhibits the lateral development induced by nitrate and regulates N/C crosstalk [22]. In rice, nitrogen deficiency and starvation strongly induced the expression of OsbZIP18 and increased BCAA levels in an OsbZIP18-dependent manner [23]. In apple, MdHY5 regulates nitrate assimilation and the transcript levels of a series of nitrate uptake genes and nitrate reductase genes [24].

Given the significant role of bZIP genes in nitrogen utilization, it is urgent to systematically characterize bZIP family members and identify the bZIP genes involved in nitrogen use in cucumber. In this study, a total of 72 CsbZIP genes were identified in the cucumber genome, and CsbZIPs could be classified into 13 groups as in Arabidopsis. In addition, chromosomal locations, gene composition, synteny analysis, and phylogenetic relationships with Arabidopsis were conducted to obtain deeper insights into CsbZIPs. To further identify the CsbZIP-mediated regulation of nitrogen utilization, RNA sequencing (RNA-seq) data and qRT–PCR were obtained, and *CsbZIP55* and *CsbZIP65* acted as candidate genes. Finally, the subcellular localization, transcriptional self-activating activity, and regulation of CsbZIP55 and CsbZIP65 to CsNPFs were preliminarily verified as regulators of nitrogen utilization in cucumber.

## 2. Results

### 2.1. Identification of bZIP Family Genes in Cucumber

To identify the bZIPs in cucumber, BLASTP (http://cucurbitgenomics.org/blast (accessed on 1 August 2022) using the AtbZIP protein sequences against the cucumber genome database (Cucumber (Chinese Long) v2 Genome) and a hidden Markov model (HMM) search using the bZIP domain (PF00170) were conducted, and a total of 72 CsbZIP genes were obtained in the cucumber genome (Appendix A). The amino acid sequences of the bZIP conserved domain from CsbZIPs were extracted and used for multiple sequence alignment. As shown in Figure 1A, the bZIP domains of CsbZIPs are composed of a basic adjacent leucine zipper structure and a DNA-binding region (Figure 1A). An invariable N-X 7-R/K motif exists in the basic DNA-binding region, and the ZIP domain contains a heptapeptide repeat of leucine (L) or related hydrophobic amino acids (Figure 1A). The highly conserved leucine residues in ZIP domains are occasionally replaced by isoleucine (Figure 1A). To analyse the classification and evolutionary relationships of CsbZIPs, a phylogenetic tree was constructed via FastTree using maximum likelihood (ML) analysis using the entire amino acid sequences of bZIP from both Arabidopsis and cucumber. As shown in Figure 1B, the CsbZIPs were clustered into 13 groups based on a study in Arabidopsis (Figure 1B; Appendix A). The S group, D group, and A group had the largest numbers of members (16, 15, and 12, respectively), while the J group, B group, and K group had only one member (Figure 1B; Appendix A). The results of multiple sequence alignment were consistent with those in Arabidopsis, indicating that the identification of CsbZIPs was convincing.

To further explore the physicochemical properties of the CsbZIPs, we analysed the amino acid sequences of CsbZIPs using ExPASy (https://web.expasy.org/ (accessed on 5 August 2022). As shown in Figure 2A and Appendix A, the isoelectric points (pI) of CsbZIPs ranged from 4.6 for CsbZIP35 to 9.83 for CsbZIP12. There were 40 acidic proteins (pI less than seven) and 15 basic proteins (pI higher than seven) in CsbZIPs (Figure 2A and Appendix A). The molecular weight (MW) of CsbZIPs ranged from 15.96 kD (CsbZIP30) to 83.47 Kd (CsbZIP25), with an average number of 35.48 kD and a difference of 19.52 kD between the maximum and minimum numbers (Figure 2B, Appendix A). The instability index of CsbZIPZIPs ranged from 37.41 (CsbZIP60) to 75.95 (CsbZIP38), and 70 of 72 CsbZIPZIPs were greater than 40, indicating that most CsbZIPZIPs were unstable proteins (Figure 2C, Appendix A). The grand average of hydropathy (GRAVY) of CsbZIPs ranged from −1.149 (CsbZIP12) to −0.229 (CsbZIP02), indicating that all CsbZIPs were hydrophilic protein proteins (Figure 1D, Appendix A).

### 2.2. Gene Structure, Domain, and Conserved Motif Analyses

To explore the evolution of the CsbZIP gene family, the exon–intron structural diversity and motif composition of each member were determined. Using MEME (https://meme-suite.org/meme/tools/meme (accessed on 6 September 2022), a total of 18 conserved motifs were identified (Figure 3A). As expected, the results of motif composition show that the CbZIP members of the same group always contained the same motifs (Figure 3A). For example, all members of the S group contain three conserved motifs. The intron number of CsbZIPs ranged from zero to thirteen. There were 15 CsbZIPs with no introns, and 7 CsbZIPs had one intron (Figure 3B). The exon numbers of the CsbZIPs ranged from one to fourteen. Sixteen CsbZIP members had only one exon, and eight CsbZIPs had two exons (Figure 3B). The intron–exon structure results show that CsbZIP members of the same group and some close members share similar gene structures (Figure 3). For example, 11 out of 16 group S members have only one exon (Figure 3B).

### 2.3. Analysis of Cis-Acting Regulatory Elements in the Promoter of CsbZIPs

The cis-acting regulatory elements (CREs) in gene promoter regions always play vital roles in the expression regulation of downstream genes, the activity of which is associated with binding with transcription factors. The promoter regions (2000 bp upstream of the translation initiation site) of CsbZIPs were obtained and used to explore CREs in the CsbZIP promoters. The CREs in the CsbZIPs were analysed using PlantCARE (https://bioinformatics.psb.ugent.be/webtools/plantcare/html/ (accessed on 15 September 2022). As shown in Figure 4, a total of 23 CREs were identified in the promoters of CsbZIPs. Among these CREs, CsbZIP promoters contained CREs related to hormones (auxin, salicylic, MeJA, abscisic acid, and gibberellin), light response (G-Box, Box 4, and MRE), drought inducibility, meristem expression, wound responsiveness, cell cycle regulation, and low temperature responsiveness (Figure 4). The CRE analysis of CsbZIPs shows that CsbZIPs might be involved in a set of biological processes.

### 2.4. Chromosomal Localization and Synteny Analysis of CsbZIP Genes

To further analyse the CsbZIP locations in the cucumber genome, TB tools were used to explore the chromosomal localization of CsbZIPs. The results of chromosomal localization analysis showed that CsbZIPs were unevenly located in the seven chromosomes of cucumber (Figure 5A). The CsbZIP gene number ranged from 5 to 16. There were 5, 11, 15, 10, 7, 16, and 8 CsbZIP genes located on chromosomes 1 to 7 (Figure 5A). The formation and expansion of gene families are always accompanied by gene duplication events. To explore the gene duplication events of CsbZIPs, BLASTP and the Multiple Collinearity Scan toolkit (MCScanX) were used. As shown in Figure 5B, there were two pairs of tandem duplication genes in the CsbZIP family, and those two tandem duplications were distributed on chromosome 6 and chromosome 7 (Figure 5B). There was a total of 22 pairs of segmental duplication genes among the CsbZIP genes distributed on seven chromosomes, and the segmental duplication events were distant (Figure 5B). The results of gene duplication events indicated that segmental duplication occurred more often than tandem duplication and that CsbZIP family expansion probably originated from tandem and segmental replication.

In addition, a comparison of syntenic maps between cucumber and Arabidopsis was conducted to further explore the phylogenetic mechanisms of the CsbZIP gene family in cucumber. A total of 66 pairs of collinear CsbZIP genes were identified between cucumber and Arabidopsis (Appendix A; Figure 5C). In cucumber, 38 out of 72 CsbZIPs in cucumber had a syntenic relationship with Arabidopsis, and 45 out of 78 bZIP genes in Arabidopsis showed a syntenic relationship with cucumber (Appendix A; Figure 5C). Among them, 20 (CsbZIP06, CsbZIP07, CsbZIP09, CsbZIP15, CsbZIP16, CsbZIP27, CsbZIP29, CsbZIP31, CsbZIP33, CsbZIP43, CsbZIP52, CsbZIP54, CsbZIP55, CsbZIP56, CsbZIP59, CsbZIP60, CsbZIP63, CsbZIP65, and CsbZIP67) out of 38 CsbZIPs had more than one homologous gene in Arabidopsis, and those CsbZIP genes might have a more important role in the evolution of the Bzip gene family in cucumber. Therefore, synteny analysis of CsbZIP genes showed that gene duplication and evolution played critical roles in bZIP gene family expansion.

### 2.5. Expression Profiles of StbZIP Genes in Different Tissues

To further explore the potential role of CsbZIPs, the transcript levels in 19 cucumber tissues were selected for characterization using previous RNA sequencing data [25]. A total of 70 of 72 CsbZIPs were identified in the RNA sequencing data, and members of CsbZIPs showed different organ-specific expression patterns in cucumber (Figure 6A, Appendix A). Among the CsbZIPs, *CsbZIP11*, *CsbZIP32*, *CsbZIP52*, *CsbZIP55*, and *CsbZIP65* were highly expressed in roots, and CsbZIP52 was specifically expressed in roots (Figure 6A). Interestingly, the expression patterns of CsbZIPs in mature leaves were similar to those in roots (Figure 6A). *CsbZIP09*, *CsbZIP28*, *CsbZIP30*, *CsbZIP35*, *CsbZIP43*, *CsbZIP62*, and *CsbZIP65* showed high expression patterns in female flowers (Figure 6A). *CsbZIP06* showed high expression in all fruit tissues (Figure 6A). The expression of CsbZIPs in female flowers and male flowers was similar, and *CsbZIP28* and *CsbZIP62* had high expression in female flowers and male flowers (Figure 6A). In fruit, the CsbZIPs in the flesh tissues had a consistent expression pattern, whereas the CsbZIP expression in the peel tissues was consistent and differed from that in flesh tissues (Figure 6A). *CsbZIP27*, *CsbZIP37*, and *CsbZIP58* had high expression in male flower buds (Figure 6A). The results of the tissue expression profile suggested that CsbZIPs had different tissue-specific expression patterns, and the CsbZIPs had similar expression patterns in some group tissues.

### 2.6. Characterization of CsbZIP Genes Involved in Nitrogen Utilization

Nitrogen utilization plays a pivotal role in plant growth and development, and previous work has shown that bZIP genes are involved in the regulation of nitrogen utilization. Our previous work indicated that most CsNPTs were downregulated by low nitrogen in leaves but upregulated in roots [16]. This regulatory mechanism of CsNPT might contribute to nitrogen utilization in roots and leaves and is still unstudied. To study the function of CsbZIP genes in the regulation of nitrogen utilization in leaves, we analysed the expression of CsbZIP genes under different nitrogen conditions using the previous RNA-sequencing data [26]. As shown in Figure 7A, *CsbZIP03* and *CsbZIP08* were upregulated in leaves under high nitrogen conditions, whereas *CsbZIP13*, *CsbZIP55*, and *CsbZIP65* were downregulated in leaves under high nitrogen conditions (Appendix A). Therefore, these five CsbZIP genes were selected for further study, and a low-nitrogen assay was conducted. As shown in Figure 7B, cucumber seedlings under low nitrogen conditions showed an obvious phenotype under nitrogen stress. In addition, we analysed the expression of CsbZIP genes in leaves under LN conditions at different time points using qRT–PCR. As shown in Figure 7C–G, the qRT–PCR results showed that *CsbZIP03* was downregulated in leaves under LN conditions, and *CsbZIP08*, *CsbZIP13*, *CsbZIP55*, and *CsbZIP65* were upregulated in leaves under LN conditions. To further verify the regulation of those five CsbZIP genes in roots under different nitrogen conditions, we analysed the CsbZIP genes in roots by qRT–PCR. Surprisingly, the qRT–PCR results showed that *CsbZIP03*, *CsbZIP08*, and *CsbZIP13* were upregulated, whereas CsbZIP55 and CsbZIP65 were downregulated in roots under LN conditions (Figure 7H–L). The contrasting regulation of *CsbZIP03*, *CsbZIP55*, and *CsbZIP65* expression indicated that those three CsbZIP genes might contribute to the regulation of CsNPF. To further analyse the expression of those five CsbZIP genes, we characterized those five CsbZIP genes in roots, stems, young leaves, mature leaves, female flowers, male flowers, and fruits at 3 and 7 days postanthesis (DPA). Consistent with the RNA sequencing data, the qRT–PCR results showed that the five CsbZIP genes were differentially expressed in diverse tissues (Figure 6B–F). Among those five CsbZIP genes, only *CsbZIP55* and *CsbZIP65* show significantly different expression in young leaves and mature leaves (Figure 6E,F). Considering the different nitrogen utilization abilities, we selected CsbZIP55 and CsbZIP65 for further study.

### 2.7. The Regulation of CsNPF Genes by CsbZIPs

To explore the regulation of CsbZIP55 and CsbZIP65 to nitrogen utilization, we attempted to characterize the regulation of CsbZIP transcription factors to CsNPF genes. We first analysed the subcellular location of the CsbZIP55 and CsbZIP65 proteins. We expressed CsbZIP55-GFP and CsbZIP65-GFP driven by the 35S promoter in *N. benthamiana*, and a green fluorescence signal was clearly observed in the nucleus of leaf epidermal cells (Figure 8A). Therefore, the CsbZIP55 and CsbZIP65 proteins were localized in the cell nucleus (Figure 8A). The results of SignalIP5.0 analysis show that approximately 70 amino acids of CsbZIP55 and CsbZIP65 were predicted as the nuclear signal peptide (Appendix A), and the result was consistent with the result of the subcellular location analysis.

In addition, we conducted transactivation activity assays to explore the transcriptional activity of CsbZIP55 and CsbZIP65. We fused the full-length CDS of CsbZIP55 (BD-CsbZIP55) and CsbZIP65 (BD-CsbZIP65) to pGBKT7 vectors and transformed BD-CsbZIP55 and BD-CsbZIP65 into AH109 yeast cells with pGADT7 vectors. The transformed yeast cells grew on SD/-Trp/-His media and turned blue on SD/-Trp/-His/+X-gal media (Figure 8B). The results of transactivation activity assays showed that CsbZIP55 and CsbZIP65 had transcriptional activation activity (Figure 8A,B).

To further explore the regulation of CsbZIP genes by CsbZIPs, we expressed CsbZIP55-GFP and CsbZIP65-GFP driven by the 35S promoter in cucumber cotyledons. We selected the CsNPF genes that were differentially expressed in leaves and roots under low and normal nitrogen conditions. The qRT–PCR results suggested that 14 out of 17 CsNPF genes were downregulated by CsbZIP55, and 6 out of 17 CsNPF genes were downregulated by CsbZIP65 (Figure 9). Therefore, CsbZIP55 and CsbZIP65 negatively regulated the expression of *CsNPFs*, which are the main transporters responsible for nitrogen utilization.

## 3. Discussion

Members of the bZIP family have been shown to have functions in nitrogen utilization. AtbZIP1, a Group S member, functions as a master regulator in nitrogen nutrient signal transduction [27]. *CsbZIP21*, the homologue of *AtbZIP1* in cucumber, has no tissue-specific expression and is not regulated by nitrogen (Figure 6A). AtTGA1 (AtbZIP47) and AtTGA4 (AtbZIP57), Group D members, are key regulators of the nitrogen response [19]. *CsbZIP13* and CsbZIP33, the homologues of *AtTGA1* and *AtTGA4* in cucumber, were highly expressed in roots and the expression of *CsbZIP13* was regulated by nitrogen in leaves and roots, indicating that CsbZIP13 and CsbZIP33 might be involved in the nitrogen response (Figure 6A,D and Figure 7E,J). AtHY5 (AtbZIP56) and AtHYH (AtbZIP64), two Group H members, are regulators of nitrate and nitrite reductase and nitrate and ammonium transporters [20,21]. *CsbZIP12* and *CsbZIP28*, the homologues of *AtHY5* and *AtHYH* in cucumber, are highly expressed in male flowers, indicating that CsbZIP12 and CsbZIP28 might be regulators of nitrogen use in male flowers (Figure 6A). AtABI5 (AtbZIP39), a Group A member, is involved in the regulation of nitrate-induced inhibition of lateral development and C/N crosstalk [22]. *CsbZIP22*, the homologue of *AtABI5*, is highly expressed in flesh tissue of fruit, indicating that CsbZIP22 might be involved in the regulation of nitrogen utilization of fruit flesh nitrogen utilization (Figure 6A). TabZIP60 increased nitrogen use and grain yield in wheat [5]. In cucumber, *CsbZIP08* is homologous to *TabZIP60* and is upregulated by nitrogen in roots and leaves (Figure 6A,C and Figure 7D,I). Therefore, CsbZIP08 might also be involved in nitrogen utilization.

Compared with previous works, CsbZIP55 and CsbZIP65 were firstly identified as regulators of nitrogen use in cucumber. Among the nitrogen-regulated CsNPF genes, most genes were regulated by CsbZIP55 and CsbZIP65. To further identify the potential target CsNPF genes of CsbZIP55 and CsbZIP65, we analysed the cis-elements in the promoters of these CsNPF genes. A previous study showed that bZIP transcription factors specifically bind to the ACGT core sequences, such as A-box (TACGTA), C-box (GACGTC), and G-box (CACGTG) [17,28]. Among the CsNPF genes regulated by CsbZIP55 and CsbZIP65, there was one A-box motif in the promoter of *CsNPF5.17*, *CsNPF5.10*, *CsNPF6.4*, and *CsNPF6.5*; there were two A-box motifs in the promoter of *CsNPF1.3*; there was one C-box motif in the promoter of CsNPF2.3; and there were one C-box and one G-box motif in the promoter of *CsNPF7.6* (Appendix A). The cis-element analysis results suggested that *CsNPF1.3*, *CsNPF2.3*, *CsNPF5.17*, *CsNPF5.10*, *CsNPF6.4*, *CsNPF6.5*, and *CsNPF7.6* are the putative target genes of CsbZIP55 and CsbZIP65.

Protein functions are often regulated by interacting proteins. The results of STRING (https://cn.string-db.org/ (accessed on 1 September 2022) showed that CsbZIP55 might interact with a number of serine/threonine protein kinases (Appendix A). CsbZIP65 might interact with HY5-like, posf21-like, abscisic acid-insensitive 5 transcription factors, and E3 ubiquitin–protein ligase herc1-like (Appendix A). Therefore, these serine/threonine protein kinases, transcription factors, and E3 ubiquitin–protein ligases might regulate the function of CsbZIP55 and CsbZIP65 and nitrogen utilization in cucumber roots and leaves.

In cucumber, nitrogen repressed the expression of CsNPF genes in leaves, whereas nitrogen actives the expression of CsNPF genes in roots [16,26]. The different regulation patterns of CsNPF genes in leaves’ and roots’ responses to nitrogen play a potential role in the nitrogen transport and utilization, and the underlying mechanisms were largely unknown. Interestingly, *CsbZIP55* and *CsbZIP65* were upregulated in leaves and downregulated in roots, respectively (Figure 7). Meanwhile, CsbZIP55 and CsbZIP65 repressed the expression of CsNPF genes (Figure 9). Those results co-suggested that CsbZIP55 and CsbZIP65 might act as the pivotal regulators of nitrogen transport and utilization in cucumber (Figure 10).

## 4. Materials and Methods

### 4.1. Plant Material and Treatment

Cucumber (*Cucumis sativus* L. “Jinyan 4”) seeds were used for gene expression analysis, gene amplification, and nitrogen treatment. Cucumber seeds were soaked in 55 °C water for 20 min and moist filter paper at 28 °C for 48 h. After seed germination, cucumber seeds were cultured with river sand (cleaned and sterilized with water in advance) in a nutrient bowl (10 cm × 10 cm) and cultured in an incubator (relative humidity: 70%, 14 h/10 h light/dark; 25 °C/18 °C day/night). The cucumber seeds at the one-leaf stage were treated with normal nitrogen and low nitrogen treatments. The nitrogen concentration of the normal nitrogen treatment was set at 3.5 mM, and the nitrogen concentration of the low nitrogen treatment was set at 0.438 mM. The nutrient solution of the normal nitrogen treatment (Ca(NO_3_)_2_·4H_2_O 826 mg/L, KNO_3_ 607 mg/L_,_ NH_4_H_2_PO_4_ 115 mg/L, and microfertilizer) was modified as the previous Hafe of Japan Yamazaki cucumber dedicated nutrient solution. The nutrient solution of the low nitrogen treatment contained the following: Ca(NO_3_)_2_·4H_2_O 103.3 mg/L, KNO_3_ 75.9 mg/L, NH_4_H_2_PO_4_ 14.4 mg/L, CaCl_2_ 339.8 mg/L, K_2_SO_4_ 365.3 mg/L, KH_2_PO_4_ 118.7 mg/L, and microfertilizer. The microfertilizer contained the following: 483 mg/L MgSO_4_·7H_2_O, 13.9 mg/L FeSO_4_, 18.6 mg/L EDTA-Na_2_, 2.86 mg/L H_3_BO_3_, 2.13 mg/L MnSO_4_, 0.22 mg/L ZnSO_4_, 0.08 mg/L CuSO_4_, and 0.02 mg/L (NH_4_)_6_Mo_7_O_24_·4H_2_O. The nutrient solutions were used for treatment once every three days. Cucumber roots were sampled after nitrogen treatment for 20 days. The river sand was removed using water, and the whole roots were frozen in liquid nitrogen and stored at −80 °C.

### 4.2. Identification of bZIP Family Members in Cucumber

The whole genome sequences of cucumber version 2 were retrieved from the Cucurbit Genomics Database (http://cucurbitgenomics.org/ftp/genome/cucumber/Chinese_long/V2 (accessed on 1 August 2022). The bZIP sequences of Arabidopsis were obtained from the Arabidopsis genome database (https://www.arabidopsis.org (accessed on 1 August 2022). First, the protein sequences of Arabidopsis bZIPs were used as queries for BLAST analysis in TBtools (e-value, 1 × 10^−10^). Then, we obtained bZIP domain (PF00170) profiles from the Pfam database (http://pfam.xfam.org/ (accessed on 3 August 2022), and the CsbZIP genes of the Cucumber Genome Database (v2.0) were identified using HMMER 3.0. The candidate sequences with incomplete or lacking domains were removed by CDD (http://www.ncbi.nlm.nih.gov/cdd/ (accessed on 3 August 2022) and Pfam (http://pfam.xfam.org/search/sequence (accessed on 3 August 2022) databases.

### 4.3. Phylogenetic Analysis and Classification of CsbZIP Genes

Phylogenetic analysis was conducted using the Arabidopsis bZIP protein sequences and the cucumber bZIP protein sequences. Based on the alignment results of the Clustal W program with the default parameters, a neighbour-joining (NJ) tree was constructed to build the phylogenetic tree. Finally, the phylogenetic tree was visualized and refined using Interactive Tree of Life (ITOL, https://itol.embl.de/itol.cgi (accessed on 1 September 2022) [29,30,31].

### 4.4. Gene Structure, Domain, and Identification of Conserved Motifs

The intron and exon structures of the CsbZIP genes were visualized using TBtools. MEME (https://meme-suite.org/meme/tools/meme (accessed on 6 September 2022) was used to analyse the conserved motifs of CsbZIP genes with the following default parameters: the maximum number of motifs was set to 10, and the optimum width of each motif was between six and 100 residues [32]. The conserved domains were analysed using Batch-CD Search with the default parameters. The above results were integrated and visualized with TBtools.

### 4.5. Cis-Regulatory Element Analysis of CsbZIP Genes

To analyse the cis-acting elements in the promoter of CsbZIP genes, 2000 bp of promoter regions upstream of the translation initiation site were extracted using TBtools and analysed using PlantCARE (http://bioinformatics.psb.ugent.be/webtools/plantcare/html/ (accessed on 15 September 2022). The cis-regulatory elements in the promoters of CsbZIP genes were visualized with TBtools [33].

### 4.6. Chromosomal Location and Collinearity Synteny Analysis

The chromosomal location information of cucumber bZIP genes was obtained from the gff3 files of the cucumber genome and visualized with TBtools. The gene duplication events and collinearity relationships of CsbZIPs were analysed using MCScanX with the default parameters [34]. For the collinearity synteny analysis, the orthologous bZIP genes from cucumber and Arabidopsis were selected for synteny relationship analysis, and collinearity maps were constructed using MCScanX and visualized using the multiple synteny plot tool in TBtools.

### 4.7. Expression Analysis of CsbZIP Genes

To explore the expression pattern of CsbZIP genes in different cucumber tissues, RNA-seq data (PRJNA312872) were acquired from the BioProject of the Cucurbit Expression Atlas, and the FPKM values of CsbZIP genes were extracted. Fifteen different tissues of cucumber, including roots, stems, young leaves, young leaf petioles, old leaves, old leaf petioles, tendrils, female flowers, male flowers, ovaries, expanded unfertilized ovaries, and expanded fertilized ovaries, were selected for transcript level analysis. The transcript levels of CsbZIP genes were visualized using the heatmap tool in TBtools. The differential CsbZIP genes that responded to nitrogen were obtained from a previous study and visualized using the heatmap tool in TBtools [26]. The primers used in this study were shown in Appendix A.

### 4.8. Subcellular Localization Assays of CsbZIP55 and CsbZIP65

The subcellular localization assays were performed according to our previous work [35]. The full coding sequences (CDSs) of CsbZIP55 and CsbZIP65 were PCR-amplified and fused with red fluorescent protein (GFP) driven by the CaMV-35S promoter into the pHellgate8 vector (named 35S: CsbZIP55-GFP and 35S: CsbZIP65-GFP). The recombinant vectors were transformed into Agrobacterium GV3101 competent cells and transiently expressed in the leaves of Nicotiana benthamiana. Three days after injection, the GFP signals were observed and captured under a confocal laser microscope (LSM 880NLO, ZEISS, Baden-Württemberg, Germany).

### 4.9. Transactivation Assays of CsbZIP55 and CsbZIP65

The full-length CDS of CsbZIP55 and CsbZIP65 were amplified and inserted into the pGBKT7 vector to fuse with a GAL4 DNA binding domain, named BD-CsbZIP55 and BD-CsbZIP65. BD-CsbZIP55 and BD-CsbZIP65 were transformed into the yeast strain AH109 using the LiAc yeast transformation system. The transformed yeasts were grown on synthetic dextrose (SD) media lacking tryptophan and histidine (SD/-Trp/-His) with or without X-gal and incubated at 30 °C for 3 days.

### 4.10. Quantitative Real-Time PCR Analysis of CsbZIP Genes

Seven cucumber tissues (roots, stems, young leaves, mature leaves, female flower, male flower, and fruit at 3 days postanthesis) were selected for tissue-specific expression. Under normal and low nitrogen conditions, the leaves and roots after 2, 6, 12, and 24 h of treatment were selected for nitrogen response analysis. Total RNA from different tissues was extracted using an RNAiso Plus kit (TaKaRa, Dalian, China). One microgram of total RNA was used for cDNA synthesis using a PrimeScript ™ RT Reagent Kit with gDNA Eraser (TaKaRa, Dalian, China). A One-Step SYBR PrimeScript RT–PCR kit (TaKaRa, Dalian, China) and CFX96 Touch™ Real-Time PCR Detection System (Bio-Rad, Berkeley, CA, USA) were used for real-time PCR analysis (qRT–PCR). The 2^−∆∆Ct^ method was applied to calculate the relative expression level. The 18S rRNA gene (Csa2G252100) was used as the internal control. Three biological replicates were performed for each experiment. Error bars in the figures represent the standard deviation of three biological replicates. The primers used for qRT–PCR are listed in Appendix A.

## 5. Conclusions

The bZIP transcription factor family is one of the largest transcription factor families in plants and is involved in plant metabolism, stress responses, and plant development [17]. Due to the vital function of bZIP in plants, bZIP gene family members have been identified in various plant species, including Arabidopsis [17], rice [36], wheat [7], tomato [37], and potato [38]. In addition, Mehmet et al. previously identified 64 bZIP family members in cucumber [39]. In this study, a total of 72 bZIP members were identified in the cucumber genome, and eight novel extra CsbZIP genes (*CsbZIP65*–*CsbZIP72*) were firstly identified. The more comprehensive analysis of CsbZIPs in this work will lay the foundation for further functional verification. The results of the expression pattern of CsbZIPs and the transient expression of CsbZIP55 and CsbZIP65 suggested that CsbZIP55 and CsbZIP65 are involved in the regulation of nitrogen use in root and leaves through NPF genes.

## Figures and Tables

**Figure 1 plants-12-03228-f001:**
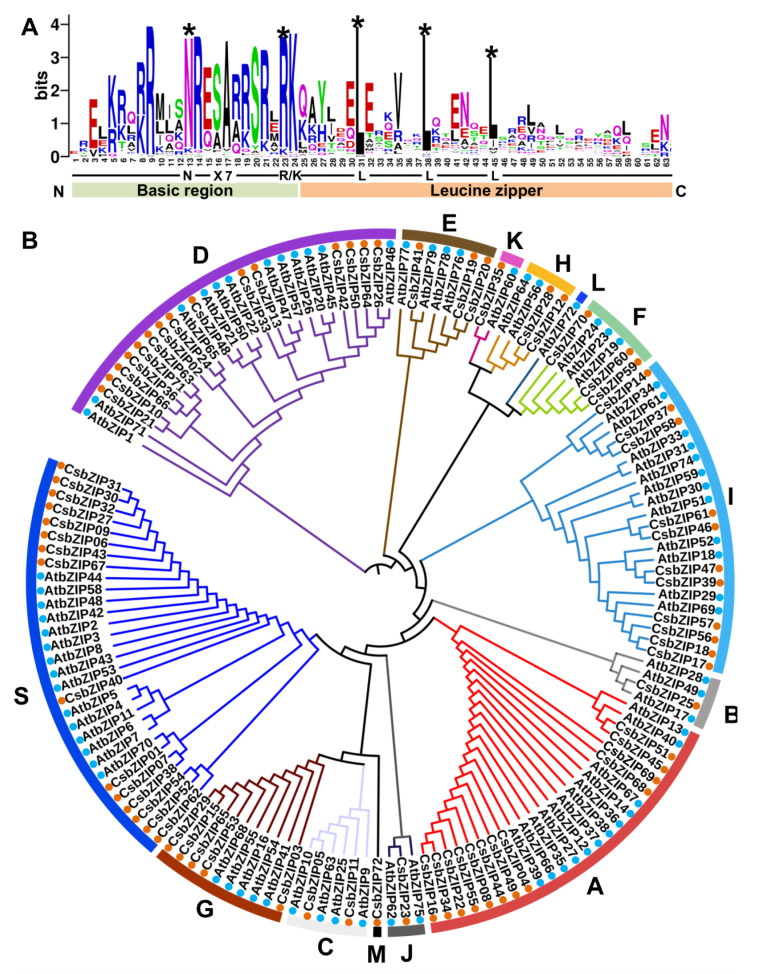
Phylogenetic analysis of the bZIP gene family in cucumber. (**A**) Visualization of multiple sequence alignment of the cucumber bZIP (CsbZIP) DNA binding domains. The x-axis represents the conserved sequences of the basic region and leucine zipper region. The total height of each letter pile indicates the conservation of each residue sequence at this position (measured in bits). The height of a single letter in the letter piles represents the relative frequency of the corresponding amino acid at that position. (**B**) Phylogenetic analysis of the bZIP gene family in cucumber and Arabidopsis. Seventy-two cucumber bZIP proteins (marked with orange circles) and 78 Arabidopsis bZIP proteins (marked with blue circles) were used for the construction of the phylogenetic tree. The clades representing different subfamilies are indicated by different colours. * marked the highest conserved amino acid; red points marked bZIP genes in Arabidopsis; blue points marked bZIP genes in cucumber.

**Figure 2 plants-12-03228-f002:**
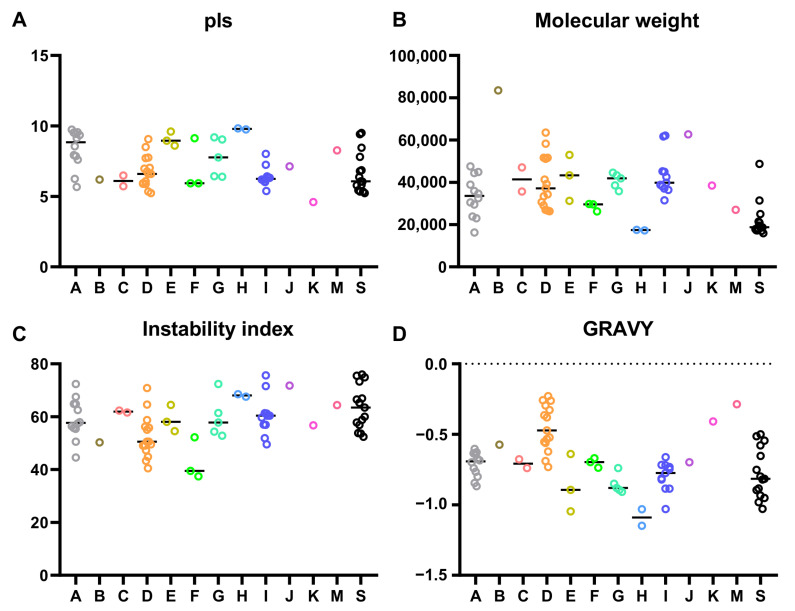
Molecular characterization of bZIP proteins in cucumber: (**A**) theoretical isoelectric point; (**B**) molecular weight; (**C**) instability index; (**D**) grand average hydropathy.

**Figure 3 plants-12-03228-f003:**
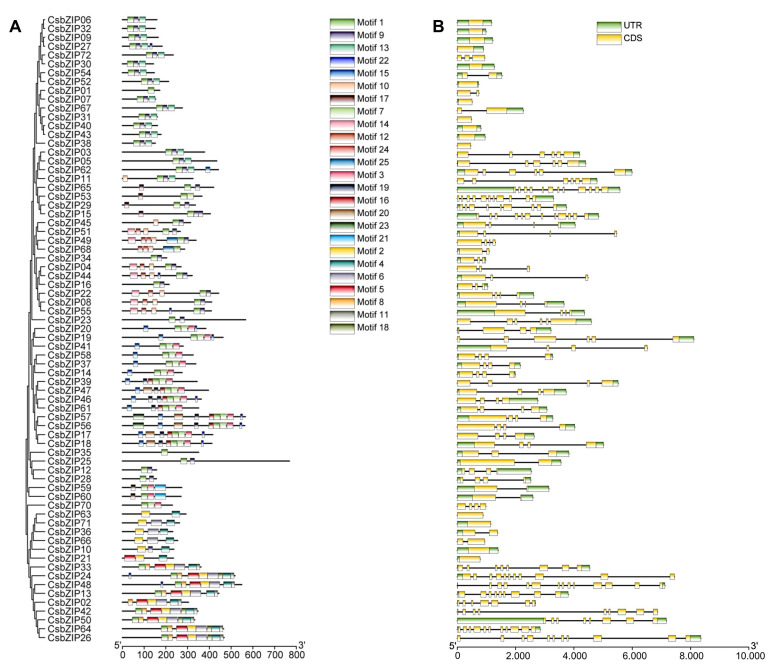
Protein motifs and DNA structures of the bZIP gene family in cucumber. (**A**) Protein motifs in the bZIP members. (**B**) Gene structure and domain analyses of bZIP genes in cucumber. Exons and 5′ UTR/3′ UTR are displayed using yellow and green boxes. The clustering represents the results of phylogenetic analysis.

**Figure 4 plants-12-03228-f004:**
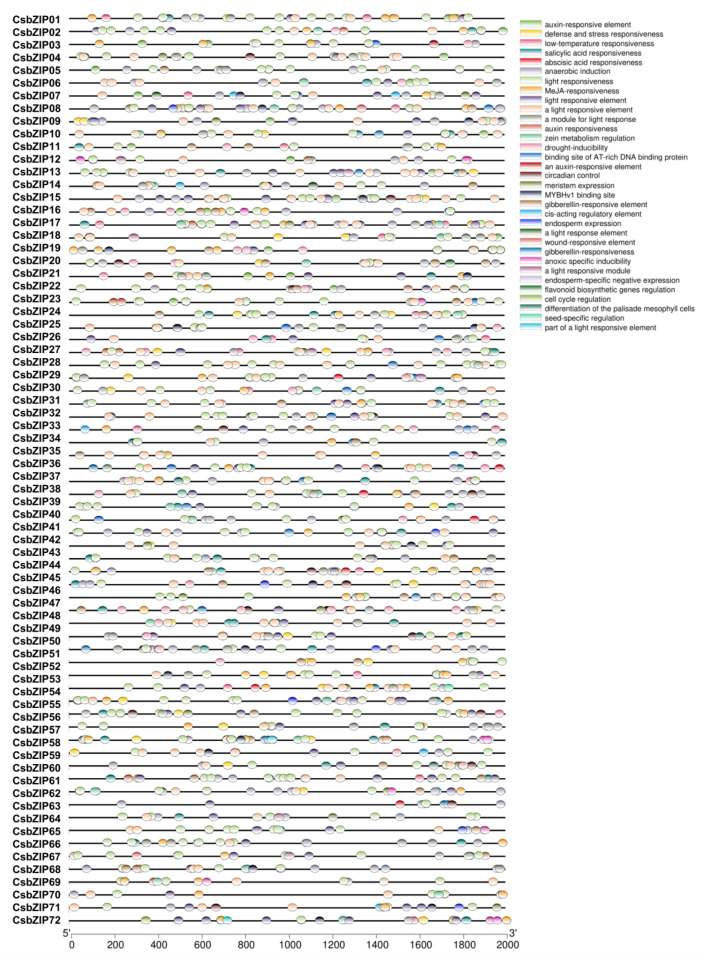
Cis-acting regulatory element (CRE) analysis in the 2000 bp promoter regions of the 72 bZIP genes in cucumber. The box fields with different colours represent different cis-acting regulatory elements predicted by MEME.

**Figure 5 plants-12-03228-f005:**
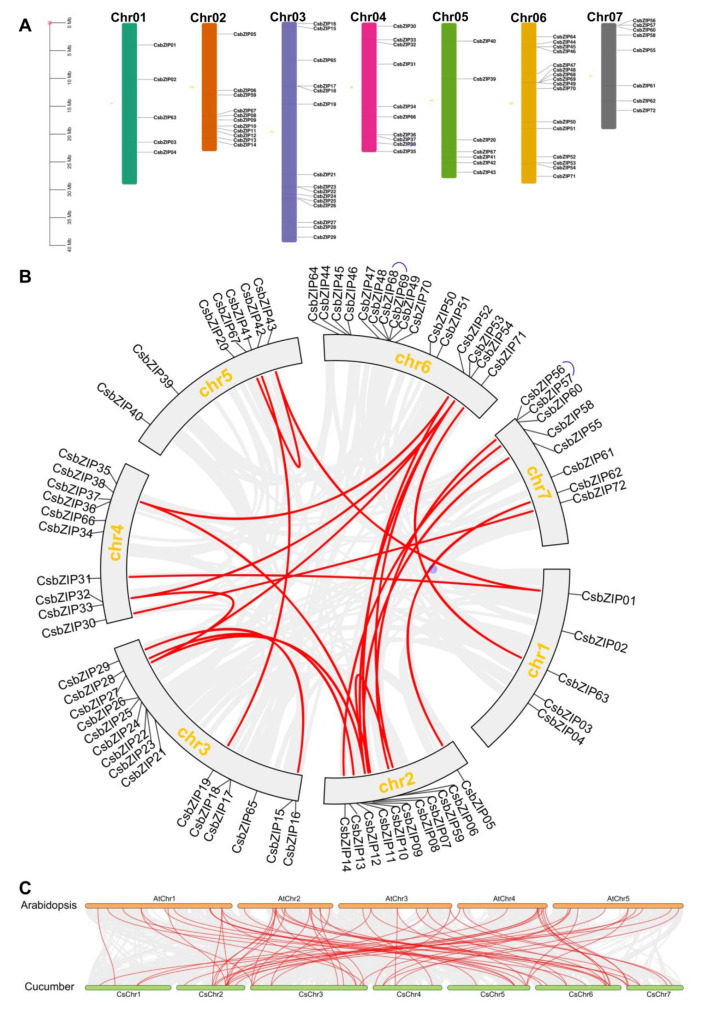
Chromosome distribution and collinearity analysis of the bZIP gene family in cucumber. (**A**) Chromosome distribution of the bZIP gene family in cucumber. (**B**) Schematic indicating the chromosomal distribution and interchromosomal relationships of bZIP genes in cucumber. Grey lines represent all syntenic blocks in the cucumber genome, purple lines in the outer ring show the tandem duplication pairs of CsbZIP genes, and red lines indicate segmental gene duplication. (**C**) Synteny analysis of NPF genes between cucumber and Arabidopsis. The grey lines indicate all collinear blocks, and the blue lines indicate collinear bZIP gene pairs between cucumber and Arabidopsis. Chr01-chr07 represent chromosomes 01–07.

**Figure 6 plants-12-03228-f006:**
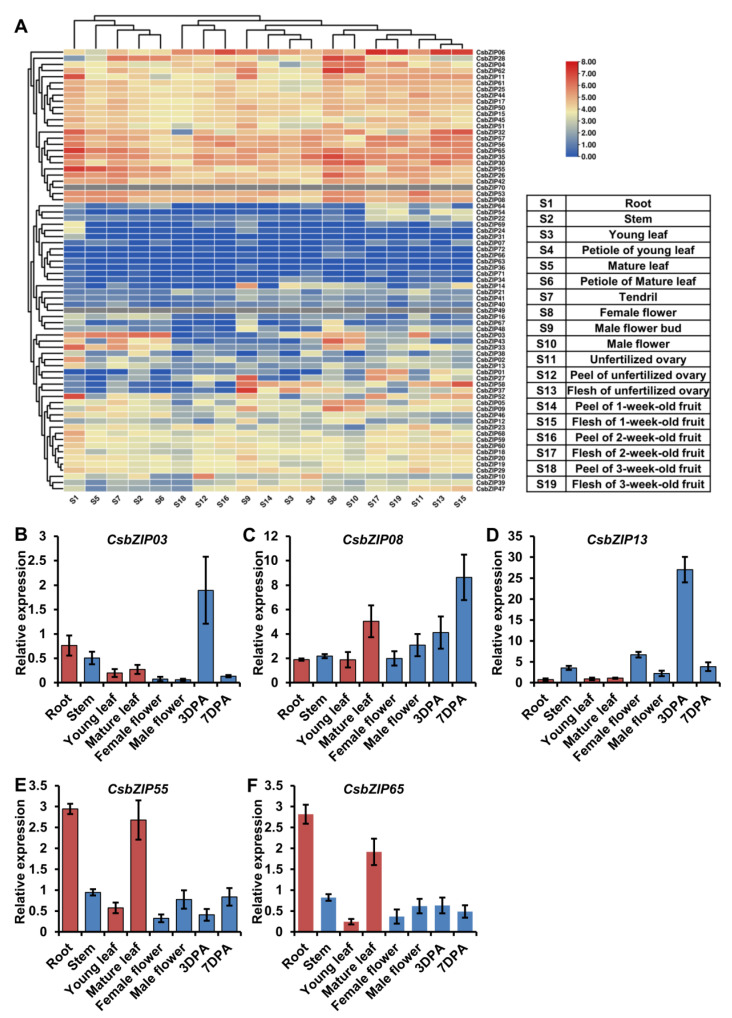
CsbZIP gene expression in different cucumber tissues. (**A**) The expression profiles of CsbZIP genes in nineteen tissues based on public transcriptome data. (**B**–**F**) CsbZIP gene expression in seven cucumber tissues using qRT–PCR; 3 DPA and 7 DPA indicate the fruit at 3 and 7 days postanthesis stages, respectively. Error bars are standard deviations from three biological replicates. Bars annotated with asterisks are significantly different according to Fisher’s least significant difference test after ANOVA. The relative expression levels were normalized to the expression of Cs18S.

**Figure 7 plants-12-03228-f007:**
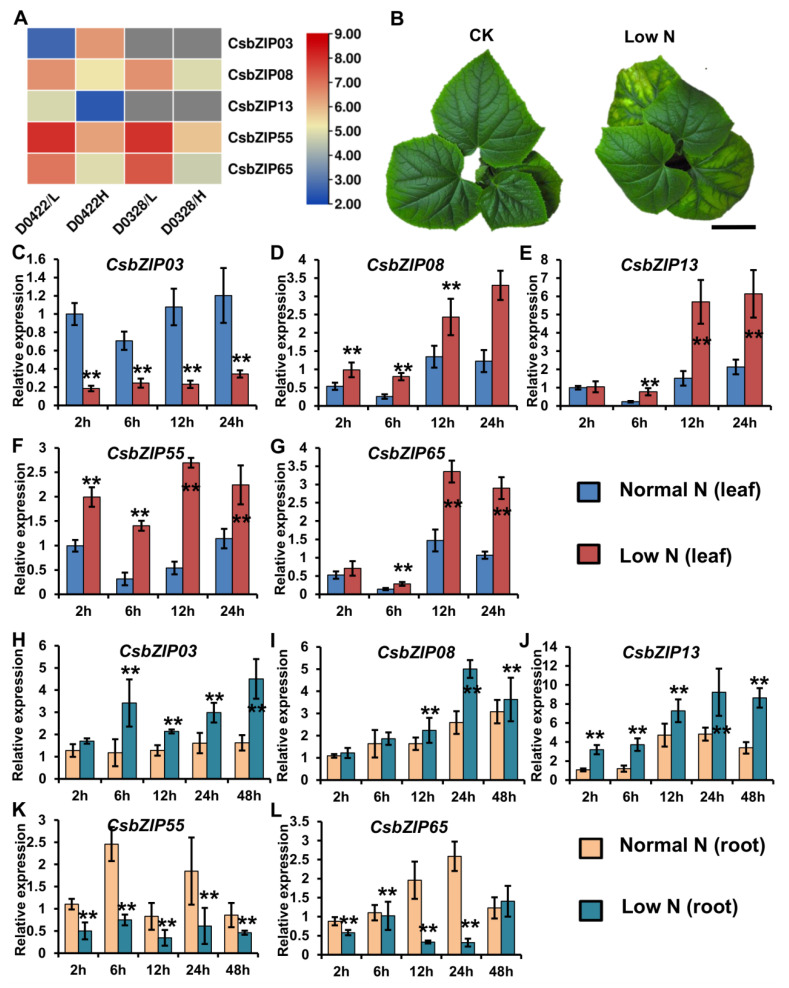
Differentially expressed bZIP genes responded to nitrogen in cucumber leaves and roots. (**A**) CsbZIP genes differentially expressed in leaves from D0328 and D0422 plants grown under low and high N conditions: D0422/L, D0422 grown under low N conditions; D0422/H, D0422 grown under high N conditions. D0328/L, D0328 grown under low N conditions; D0328_H, D0328 grown under high N conditions. (**B**) Cucumber plants under normal nitrogen and low nitrogen conditions. (**C**–**G**) CsbZIP gene expression in leaves under normal nitrogen and low nitrogen conditions. (**H**–**L**) CsbZIP gene expression in roots under normal nitrogen and low nitrogen conditions. Error bars are standard deviations from three biological replicates. Bars annotated with asterisks are significantly different according to Fisher’s least significant difference test after ANOVA (**, *p* < 0.01). The relative expression levels were normalized to the expression of Cs18S.

**Figure 8 plants-12-03228-f008:**
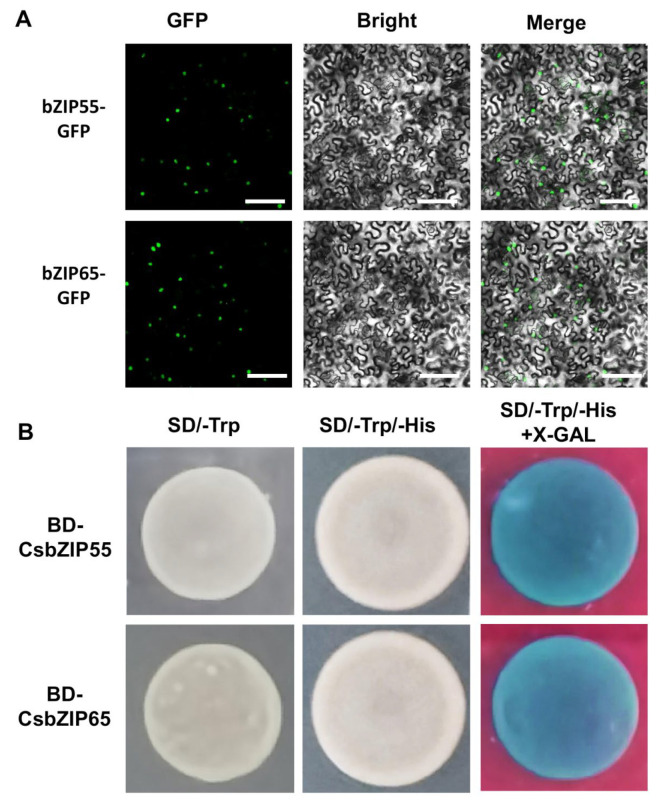
Subcellular localization and transactivation analysis of CsbZIP55 and CsbZIP65. (**A**) The 35S:CsbZIP55-GFP and 35S:CsbZIP65-GFPs were located in the nucleus of tobacco leaves. green dot indicated nuclear. Bars = 500 μm. (**B**) Transactivation activity assay of CsbZIP55 and CsbZIP65 in yeast strain AH109. β-galactosidase activities against X-gal were detected on SD/-Trp/-His media.

**Figure 9 plants-12-03228-f009:**
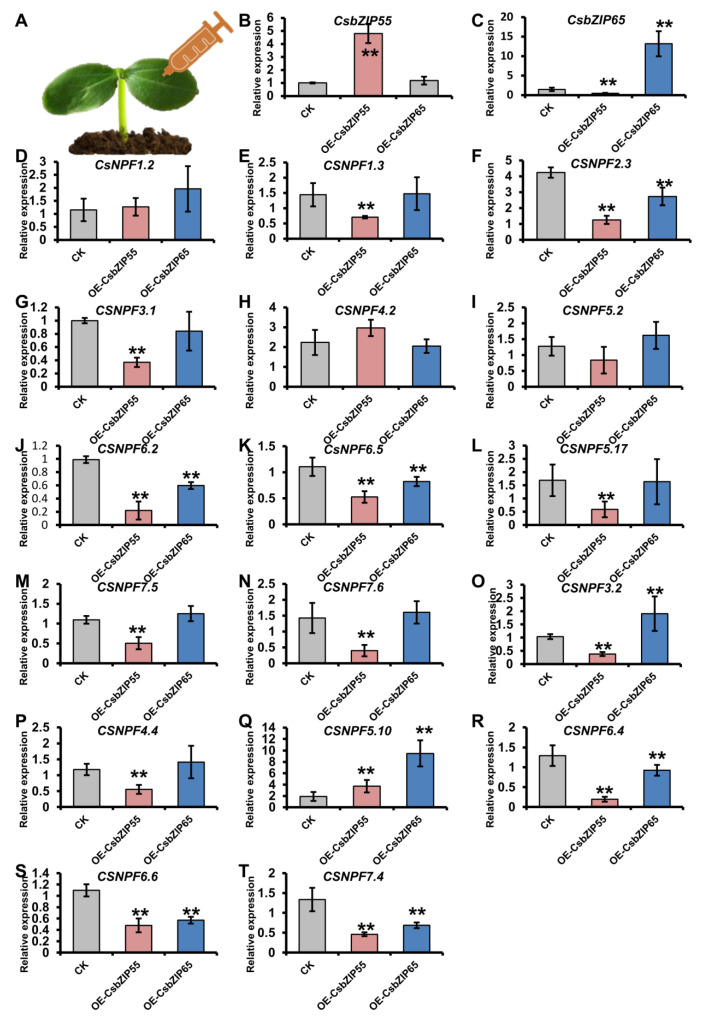
CsbZIP55 and CsbZIP65 regulate the expression of CsNPF genes. (**A**) Schematic representation of transient expression of CsbZIP55 and CsbZIP65 in cucumber cotyledons. Gene expression was chartered at five days after injection. (**B**,**C**) The relative expression of CsbZIP55 (**B**) and CsbZIP65 (**C**) in plants transiently expressing 35S: CsbZIP55-GFP and 35S: CsbZIP65-GFP. (**D**–**T**) The relative expression of CsNPF genes in the plants transiently expressing 35S: CsbZIP55-GFP and 35S: CsbZIP65-GFP. Bars annotated with asterisks are significantly different according to Fisher’s least significant difference test after ANOVA (**, *p* < 0.01). The relative expression levels were normalized to the expression of Cs18S.

**Figure 10 plants-12-03228-f010:**
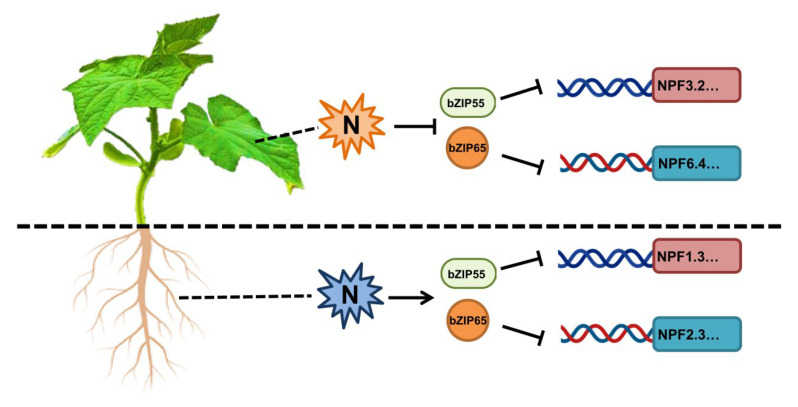
Proposed model for the different regulation of CsbZIP55 and CsbZIP65 to CsNPF genes in cucumber roots and leaves. In cucumber leaves, nitrogen signalling represses the expression of CsbZIP55 and CsbZIP65 and activates the expression of CsNPF genes. In contrast, nitrogen signalling activates the expression of CsbZIP55 and CsbZIP65 and represses the expression of CsNPF genes in roots.

## Data Availability

Not applicable.

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
