# Peer review of "The Potential Role of bZIP55/65 in Nitrogen Uptake and Utilization in Cucumber Is Revealed via bZIP Gene Family Characterization"

_plants, 2023, doi:10.3390/plants12183228_

Round 1
Reviewer 1 Report
The manuscript entitled “Genome-wide identification and characterization of the bZIP gene family reveal the potential regulation of bZIP55/65 in nitrogen absorption and distribution in cucumber (Cucumis sativus L.)” described the role of this TF gene family in Nitrogen use efficiency in cucumber.
I found the manuscript well organized and the results are consistent with the aim of the proposed research. I have only two minor suggestions:
1. It is necessary a moderate revision of the English style along to all the manuscript sections
2. I would suggest to simplify the title of the manuscript with the addition of some new keywords, based on the cutted word from the title; I could suggest this title “The potential role of bZIP55/65 in nitrogen uptake and utilization in cucumber is revealed by the bZIP gene family characterization”.
3. Please, verify that the addresses of the Authors were not repeated.
Once the Authors have addressed these two minor revision requests, I will retain the manuscript suitable for publication in Plants – MDPI.
As I wrote in the other section of the revision, the manuscript needs a moderate revision of the English style.
Author Response
The manuscript entitled “Genome-wide identification and characterization of the bZIP gene family reveal the potential regulation of bZIP55/65 in nitrogen absorption and distribution in cucumber (Cucumis sativus L.)” described the role of this TF gene family in Nitrogen use efficiency in cucumber.
I found the manuscript well organized and the results are consistent with the aim of the proposed research. I have only two minor suggestions:
- It is necessary a moderate revision of the English style along to all the manuscript sections.
Response: Thanks for the suggestion. We have revised the English style along to all the manuscript sections.
- I would suggest to simplify the title of the manuscript with the addition of some new keywords, based on the cutted word from the title; I could suggest this title “The potential role of bZIP55/65 in nitrogen uptake and utilization in cucumber is revealed by the bZIP gene family characterization”.
Response: Thanks for the enlightening comments. We have revised the title as “The potential role of bZIP55/65 in nitrogen uptake and utilization in cucumber is revealed by the bZIP gene family characterization”.
- Please, verify that the addresses of the Authors were not repeated.
Response: Thanks for the suggestion, and we have checked the addresses of the Authors as required.
Once the Authors have addressed these two minor revision requests, I will retain the manuscript suitable for publication in Plants – MDPI.
Reviewer 2 Report
The Introduction does not contain all the essential information and requires rephrasing. The aim of the study needs correction as it is too lengthy. The 'Materials and Methods' section is significantly misplaced after the research results. The elements of 'Materials and Methods' are found in the 'Results' section, which should not be the case in an original scientific paper. Figure 3 is not very clear. On what basis was the proposed model for the different regulation of CsbZIP55 and CsbZIP65 to CsNPF 393 genes in cucumber roots and leaves prepared? On what basis were specific nitrogen concentrations (normal, low) used in the experiment (Wang et al., 2022)? The 'Conclusions' section merely summarizes the research findings. There is a notable lack of discussion in the paper. Once these issues are addressed and corrected, the paper should be ready for publication.
Author Response
The Introduction does not contain all the essential information and requires rephrasing. The aim of the study needs correction as it is too lengthy. The 'Materials and Methods' section is significantly misplaced after the research results.
1.The elements of 'Materials and Methods' are found in the 'Results' section, which should not be the case in an original scientific paper.
Response: Thanks for the comments. We have removed the elements of 'Materials and Methods' in the 'Results' section.
2.Figure 3 is not very clear.
Response: Thanks for the comments. We have replaced the Figure 3 in the revised manuscript.
3.On what basis was the proposed model for the different regulation of CsbZIP55 and CsbZIP65 to CsNPF genes in cucumber roots and leaves prepared?
Response: Thanks for the comments. To explore the response of CsbZIP55 and CsbZIP65 to nitrogen, we analyzed the expression of CsbZIP55 and CsbZIP65 in roots and leaves. The results of qRT-PCR show that CsbZIP55 and CsbZIP65 were up-regulated in leaves, whereas CsbZIP55 and CsbZIP65 were downregulated in roots under Low-Nitrogen conditions. Further, we transient expressed CsbZIP55 and CsbZIP65 in cotyledon, and the several CsNPF genes putative involved in the nitrogen use were down-regulated. Hence, CsbZIP55 and CsbZIP65 might act as repressor in leaves and activator of roots of nitrogen transport.
4.On what basis were specific nitrogen concentrations (normal, low) used in the experiment (Wang et al., 2022)?
Response: Thanks for the comments. We conducted nitrogen treatment based on the previous study (Qu wt al., 2019). The nitrogen concentrations (1.75mM) in low nitrogen treatment were 1/8 of normal nitrogen treatment (14mM). The
- The 'Conclusions' section merely summarizes the research findings. There is a notable lack of discussion in the paper.
Response: Thanks for the comments. Based on the previous discussion, we added the section as below. In cucumber, nitrogen repressed the expression of CsNPF genes in leaves, whereas nitrogen actives the expression of CsNPF genes in roots (Zhang et al., 2023; Du et al., 2018). The different regulation pattern of CsNPF genes in leaves and roots response to nitrogen play the potential role in the nitrogen transport and utilize, and the underlying mechanism were largely unknown. Interesting, CsbZIP55 and CsbZIP65 were up-regulated in leaves and down-regulated in roots, respectively (Figure 7). Meantime, CsbZIP55 and CsbZIP65 repressed the expression of CsNPF genes (Figure 9). Those results co-suggested that CsbZIP55 and CsbZIP65 might acted as the pivotal regulators of nitrogen transport and utilize in cucumber.
Once these issues are addressed and corrected, the paper should be ready for publication.

Reviewer 3 Report
plants-2528558
Hua et al presented a manuscript entitled “Genome-wide identification and characterization of the bZIP gene family reveal the potential regulation of bZIP55/65 in nitrogen absorption and distribution in cucumber (Cucumis sativus L.) “for publication consideration in Plants.
This manuscript reported on molecular characterization of 72 bZIP genes (CsbZIPs) in the cucumber genome which could be classified into 13 groups encoding proteins play crucial roles in various biological functions. They studied how these genes function with nitrogen in cucumber (Cucumis sativus) in its shallow roots.
The manuscript is well organized and presented. Background introduction is sufficient, results and discussion sections are sound. Its scientific soundness of this manuscript is acceptable. It meets the aims and scope of Plants journal.
This study is important and interesting. I believe this manuscript should be considered for publication with Plants journal after addressing minor issue such as formatting, Authors & affiliations were not following journal’s instruction.
Author Response
This manuscript reported on molecular characterization of 72 bZIP genes (CsbZIPs) in the cucumber genome which could be classified into 13 groups encoding proteins play crucial roles in various biological functions. They studied how these genes function with nitrogen in cucumber (Cucumis sativus) in its shallow roots.
The manuscript is well organized and presented. Background introduction is sufficient, results and discussion sections are sound. Its scientific soundness of this manuscript is acceptable. It meets the aims and scope of Plants journal.
This study is important and interesting. I believe this manuscript should be considered for publication with Plants journal after addressing minor issue such as formatting, Authors & affiliations were not following journal’s instruction.
Response: Thanks for the comments. We have addressed formatting, Authors & affiliations accordingly.